

# GenACO a multi-objective cached data offloading optimization based on genetic algorithm and ant colony optimization

Mulki Indana Zulfa[1,2], Rudy Hartanto[1], Adhistya Erna Permanasari[1] and Waleed Ali[3]

[1] Department of Electrical and Information Engineering, Universitas Gadjah Mada, Yogyakarta, Special Region of Yogyakarta, Indonesia
[2] Department of Electrical Engineering, Universitas Jenderal Soedirman, Purwokerto, Central Java, Indonesia
[3] Department of Information Technology, King Abdul Aziz University, Jeddah, Kingdom of Saudi Arabia

Corresponding author
Rudy Hartanto, rudy@ugm.ac.id

## ABSTRACT

**Background**. Data exchange and management have been observed to be improving with the rapid growth of 5G technology, edge computing, and the Internet of Things (IoT). Moreover, edge computing is expected to quickly serve extensive and massive data requests despite its limited storage capacity. Such a situation needs data caching and offloading capabilities for proper distribution to users. These capabilities also need to be optimized due to the experience constraints, such as data priority determination, limited storage, and execution time.

**Methods**. We proposed a novel framework called Genetic and Ant Colony Optimization (GenACO) to improve the performance of the cached data optimization implemented in previous research by providing a more optimum objective function value. GenACO improves the solution selection probability mechanism to ensure a more reliable balancing of the exploration and exploitation process involved in finding solutions. Moreover, the GenACO has two modes: cyclic and non-cyclic, confirmed to have the ability to increase the optimal cached data solution, improve average solution quality, and reduce the total time consumption from the previous research results.

**Result**. The experimental results demonstrated that the proposed GenACO outperformed the previous work by minimizing the objective function of cached data optimization from 0.4374 to 0.4350 and reducing the time consumption by up to 47%.

# INTRODUCTION

The conduct of work and school activities from home is the new habit adopted during the COVID-19 pandemic *Nimrod (2020)*. It led to a massive surge in internet and digital technology users (*De', Pandey & Pal, 2020*). Government and business owners are also required to maximize websites and social media to disseminate information *Haman (2020)*, while an increase was also experienced in the use of e-learning and e-banking platforms (*Azlan et al., 2020*; *Naeem & Ozuem, 2021*). Meanwhile, data exchange and management have been observed to be improving with the rapid growth of 5G technology,

edge computing, and the Internet of Things (IoT) (*Sai, Fan & Fan, 2020*; *Zhang, 2020*). Moreover, edge computing is expected to quickly serve extensive and massive data requests despite its limited storage capacity *Wang et al. (2019)*. Such a situation needs data caching and offloading capabilities for proper distribution to users *Shi et al. (2016)*. These capabilities also need to be optimized due to several constraints the experience, such as data priority determination and limited storage and execution time *Bala & Chishti (2019)*.

The data offloading mechanism is also used in a web caching strategy known as cache replacement (*Ali, Shamsuddin & Ismail, 2011*) with the focus on cached data management strategies to ensure faster data requests through cache servers (*Ali, Mariyam & Samad, 2012*; *Ali, Shamsuddin & Ismail, 2012*). Moreover, direct data requests on the origin server cloud are required to be suppressed as possible to reduce the communication latency on the user side and bandwidth efficiency on the server side (*Mertz & Nunes, 2017*; *Mertz & Nunes, 2018*). However, the data offloading process in this cache replacement also has the same main problem: determining cached data priority due to the limited cache server capacity (*Tatar et al., 2014*).

The development of 5G networks, IoT, and intelligent devices requires a reliable edge network for data distribution and interconnection to maintain Quality of Service (QoS) (*Prerna, Tekchandani & Kumar, 2020*; *Carvalho et al., 2020*). A reliable data offloading on the edge network can save 30%–40% energy and maximize caching system performance (*Shi et al., 2016*; *Ha et al., 2014*). Figure 1 illustrates the data offloading process in a fog computing environment with data priority and server capacity issues. Some of the data is not included in the edge network caching system due to limited storage capacity. Therefore, the edge network should assess the priority of cached data to minimize data service requests made to the origin server on the cloud network. For example, which data should be stored in the caching system (Z) when the storage capacity on the network edge is 1200 MB? Z = {1,2,3} or Z = {1,2,4} or Z = {3,4,5} or Z = {2,5} or a combination of other data. It is important to note that the selected cached data should have the maximum profit.

In previous studies, several methods such as Long Short Term Memory (LSTM) (*Pescosolido, Conti & Passarella, 2019*), Reinforcement Learning (RL) *Huang et al. (2019)*, and Branch and Bound (BnB) (*Elgendy et al., 2019*) have been used to solve problems in optimizing data offloading. However, these methods still leave issues such as being easily trapped at the local optimum, high time complexity, and not focusing on the priority of cached data. The utilization of the server cache is not optimal. Such a situation can cause the edge server to request the same data repeatedly to the origin server. Therefore, *Wang et al. (2019)* conducted a study on cached data offloading optimization based on Swarm Intelligence (SI) using the ACO-GA hybrid algorithm. The SI method effectively solves optimization problems because of its reliability in finding the (near) optimum solution *Ertenlice & Kalayci (2018)*.

The Cyclic ACO-GA proposed by *Wang et al. (2019)* can solve the cached data offloading problem. However, the dominant iteration issued to ACO makes the execution time quite long. In addition, the mechanism for selecting a solution using a roulette wheel makes the resulting solution does not converge quickly. Moreover, the iteration restrictions on the

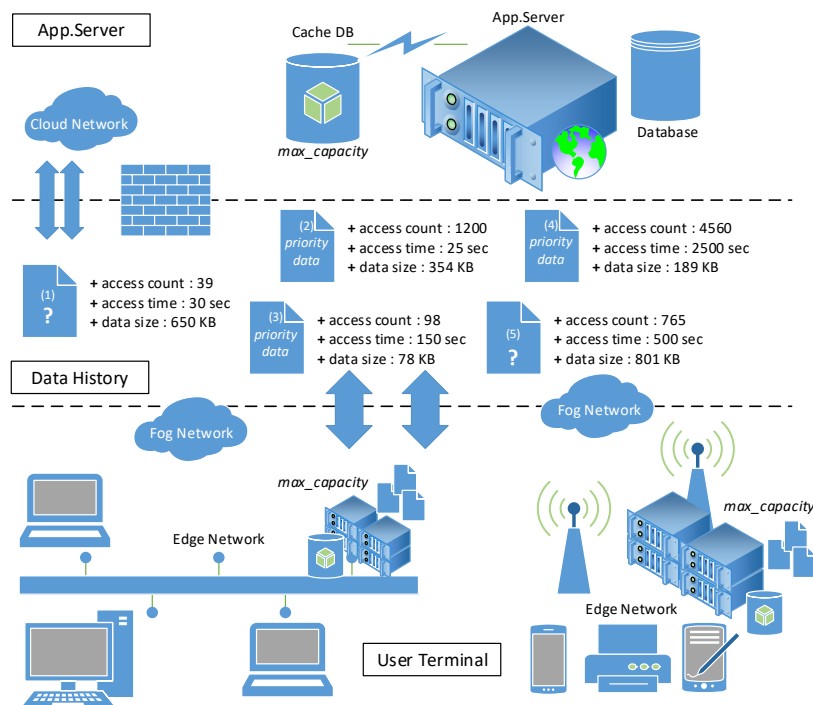

**Figure 1  Fog computing environment.**

GA algorithm make GA performance not optimal. Hence, the main research question is: "How can we hybrid and enhance the performance of evolutionary algorithms to optimize the cached data offloading ?"

Therefore in this study, we propose a new GenACO framework to improve Cyclic Genetic Ant Colony (CGACA). This paper proposed a hybrid method of the ACO-GA algorithm to solve the cached data offloading problem. The contributions are: (i) improving solution probability calculation on cached data offloading: (ii) proposing a novel framework named GenACO with cyclic and non-cyclic modes to improve previous research using more optimal profit cached data.

The following section discusses the previous related works on cached data offloading. The algorithms section comprehensively describes the working principle of ACO, GA, and CGACA framework algorithms and their performance in this research. Moreover, the methodology section discusses the proposed performance improvement of the ACO-GA hybrid as tested on a novel GenACO framework with cyclic and non-cyclic modes, followed by the dataset, simulation setup, results, and discussion sections and concluded.

## Related works

Cached data offloading is an extraordinary capability required to be owned by an application, workstation, server, and network to manage data storage considered to be larger than its capacity (*Zulfa, Hartanto & Permanasari, 2020*). It is a very common method used in cloud computing (*Wang et al., 2019*; *Li et al., 2020a*; *Li et al., 2020b*),

mobile computing (*Dutta & Vandermeer, 2017*; *Zhu & Reddi, 2017*), operating system (*Silberschatz, Galvin & Gagne, 2008*; *Tian & Liebelt, 2014*), and telecommunication *Prerna, Tekchandani & Kumar (2020)*.

*Luo et al. (2017)* examined the energy consumption optimization in Mobile Edge Computing (MEC) by formulating objective functions based on the variables of energy consumption, backhaul capacities, and content popularity. The research utilizes the GA algorithm with the optimization function to minimize the energy consumption value and validated by measuring the system average delay and average power toward adding the MEC servers and backhaul capacity. Moreover, *Xu et al. (2019)* also proposed a data offloading optimization framework named COM, designed to optimize mobile devices execution time and energy consumption. The COM framework was used to model a multi-objective optimization solved using the NSGA-III algorithm. It was validated by calculating the maximum value of the utility and resource utilization functions achieved.

*Pescosolido, Conti & Passarella (2019)* developed a Content Delivery Management System (CDMS) method for device-to-device data offloading in cellular networks. It was proposed to predict system performance in transmission distance and energy consumption using simulations configured with Infrastructure-to-Device (I2D) transmission and Base Stations. The research was validated through the efficiency of data offloading and Physical Resource Block (PRB). Furthermore, *Zhao et al. (2019)* proposed mobile data offloading by predicting the real-time traffic Small Base Station (SBS) on MEC using a multi-Long Short-Term Memory (LSTM) algorithm. The results predicted were further used for data offloading by utilizing the Cross-Entropy (CE) algorithm. The research formulated the problem with an optimization approach to determine the maximum value from the system throughput.

*Huang et al. (2019)* also developed the Deep-Q Network (DQN) framework to solve data offloading optimization and resource allocation in MEC using energy costs, computation costs, and delay costs as variables. The optimization was modeled with mixed-integer non-linear programming solved by Reinforcement Learning (RL) algorithm, and the DQN was validated by calculating the minimum total cost generated. Another research by *Elgendy et al. (2019)* focused on resource allocation optimization and computation offloading with additional data security protection in MEC. The security protection was conducted by adding the Advanced Encryption Standard (AES) algorithm to prevent cyber-attacks. At the same time, the optimization problem was modeled based on the knapsack problem and solved using a branch and bound algorithm. Its implementation was further validated by determining the smallest value of time and energy consumption.

*Kuang et al. (2021)* modeled data offloading and resource allocation in one cooperative scheme to optimize power allocation and CPU cycle frequency in MEC and, subsequently, to make appropriate data offloading decisions. The research was measured by calculating the smallest possible task latency value and also adopted the Convex optimization method, dual Lagrangian decomposition, and ShenJing Formula. Another study by *Zhong et al. (2021)* discussed a caching strategy framework to optimize traffic load, and QoS in Multi-access Edge Computing named GenCOSCO. The aim was to minimize the task execution time as a Mixed Integer Non-Linear Programming (MINLP) optimization problem by considering

the heterogeneity of task requests, pre-storage of application data, and cooperation of the base station variables. The GenCOSCO was used to propose the FixCS algorithm and was validated by calculating the average latency.

Moreover, *Peng et al. (2021)* proposed an application paradigm as a service chain for detailed data offloading and location caching mechanisms. Each service chain was limited by leasing costs and designed according to the queuing theory in computer networks. The research was further validated by calculating the average response delay towards increasing cache server time and capacity.

*Wang et al. (2019)* also studied the cached data offloading optimization on the edge network to reduce requests made to the origin server for the same data. The focus was on the cache capacity on the limited edge network, which means each cached data needed to be prioritized. The optimization problem was proposed using a knapsack problem. This research used cyclic ACO-GA with three variables: access count, access time, and data size. Research by *Wang et al. (2019)* is used as the baseline in our study because, based on our manual calculations, there is an opportunity to improve the value of the formulated objective function. The previously generated objective function value is 0.4374, but this value can be even more optimal up to 0.4350. This is a strong basis for us to improve the performance of CGACA.

## EVOLUTIONARY ALGORITHMS

### Genetic algorithm (GA)

GA was first introduced by John Holland in a publication entitled Adaptation in Natural and Artificial Systems *Holland (1992)* and observed to have adopted several phenomena such as natural selection and mutation for survival. GA used these adaptation principles to improve solutions in each generation *Purnomo (2014)*. One other main principle is a crossover, a cross-breeding mechanism between two individuals (parents) to produce better quality offspring than both parents. It illustrates in Fig. 2, where two individual chromosomes (parents) create new offspring working in groups with other genes to form a chromosome. This chromosome represents a solution vector from the completed case study, with each gene can mutate based on a certain probability, as illustrated in Fig. 3. Meanwhile, the quality of the genetic algorithm solution vector was measured by a fitness value. The highest value among the existing chromosomes is selected when the genetic algorithm finds the maximum optimization value. This fitness value was, calculated using the following Eq. (1). GA also uses elitism to maintain one or several of the best individuals in the next generation to produce other individuals with better fitness values *Santosa & Ai (2017)*.

$$fitness = \frac{1}{f_{(x)} + \varepsilon} \tag{1}$$

### Ant colony optimization (ACO)

Dorigo first introduced ACO to find the shortest path in the Travelling Salesman Problem case study (TSP) (*Dorigo, Maniezzo & Colorni, 1996*). This algorithm adopts the behavior

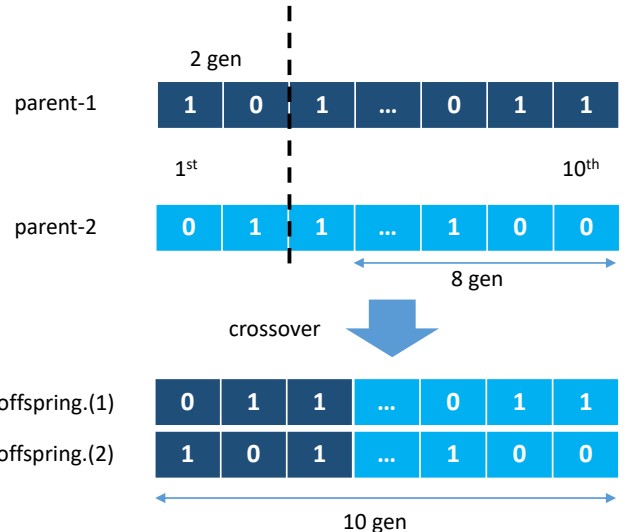

**Figure 2** Crossover mechanism of genetic algorithm.

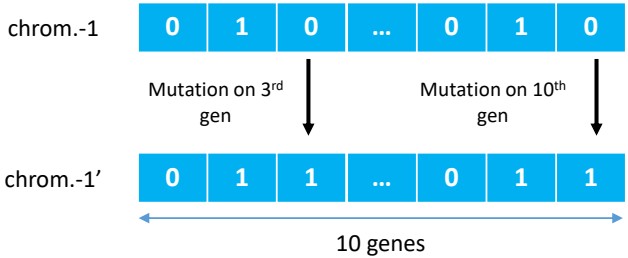

**Figure 3** Examples of mutations of genetic algorithm.

of ant colonies from the nest to the food source. Each ant can leave a pheromone trail on each of its paths. The pheromone substances are from endocrine glands, which can identify fellow ants or their groups by serving as a solid signal to influence other ants to follow in the footsteps of this pheromone. At first, each ant can determine its route and leave a pheromone trail to identify others, as indicated in Fig. 4.

Moreover, the pheromone can evaporate, which means its path is lost faster on longer routes than shorter ones. It is difficult for other ants to follow the path on a longer route due to the evaporation of the pheromone trail. In comparison, the shorter route still has traces even though the first pheromone has disappeared, but it regains the path through the pheromone from other ants. The stronger signal attracts other ants to follow the same trail, as shown in Fig. 5. An ant k at node r can select a route s(i,j) based on a certain probability, and those that have completed a route leaves a pheromone trail ($\tau$) of $\tau_{i,j} \leftarrow \tau_{i,j} + \Delta\tau^k$ while

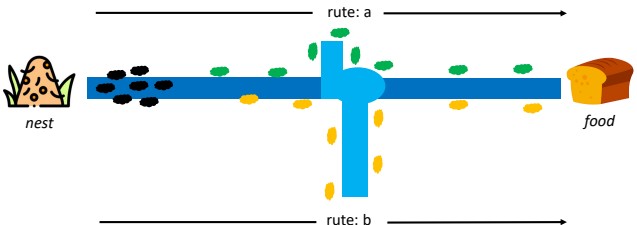

**Figure 4** Initial conditions between ant colony in a nest and food source (*Dorigo, Maniezzo & Colorni, 1996*).

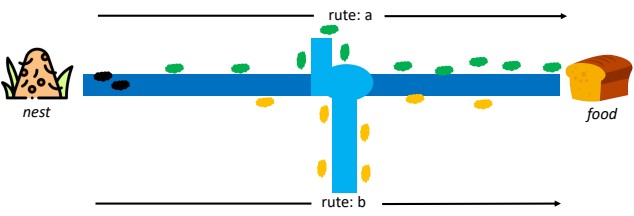

**Figure 5** Rute-a as the shortest path from a nest to food source by ant colony (*Dorigo, Maniezzo & Colorni, 1996*).

the vaporized pheromone is calculated using $\tau_{i,j} \leftarrow (1 - \rho)\,\tau_{i,j}$ where $\rho$ is a predetermined constant.

## Cyclic ACO-GA

The performance of the Cyclic ACO-GA (CGACA) proposed by *Wang et al. (2019)* was improvised in this study to improve its final result by obtaining a more optimal objective function value. It is important to note beforehand that CGACA conducts cyclical algorithmic exchange between ACO and GA with the ACO algorithm run in the first iteration. At the same time, the next solution search process is implemented through GA. ACO is applied when GA experiences stagnation in five successive iterations until the maximum iteration is achieved. Moreover, The ACO algorithm in CGACA used the roulette wheel, which has an unfavorable impact due to its production of random solutions and superior individuals (*Moodi, Ghazvini & Moodi, 2021*; *Lipowski & Lipowska, 2012*). However, the CGACA framework proposed by *Wang et al. (2019)* does not provide settlement action in a situation where the GA does not experience stagnancy. It means the GA does not have a full role in exploiting the solution to the fullest.

The results of our initial research showed that the ACO algorithm takes the longest time when compared to GA and binary Particle Swarm Optimization (BPSO) in completing the cached data optimization. The worst-case scenario in CGACA will occur if the GA algorithm experiences a solution stagnation at the beginning of the iteration so that ACO will be rerun until the iteration is complete. Such a situation makes the ACO iteration dominate, so the total CGACA time consumption becomes very large. Table 1 explains the important weakness points to be considered in improving the CGACA performance.

**Table 1  The weaknesses of CGACA framework.**

| No | Aspect | Explanation of weakness |
|---|---|---|
| 1 | Time consuming | In worst case condition, CGACA takes a very long time because the dominant iteration is held by ACO. |
| 2 | Dominant iteration | The solution search process is completed by ACO over a very long time when the GA experiences a solution stagnation condition at the beginning of the iteration. |
| 3 | GA iteration limitation | The CGACA framework limits the GA to a maximum of 20 iterations from a total of 100 allowed for the ACO. This limits the optimal performance of the GA. |
| 4 | Solution probability | The ACO algorithm in the CGACA framework uses a roulette wheel mechanism for solution selection and this has an unfavorable impact due to its production of random solutions and superior individuals (*Moodi, Ghazvini & Moodi, 2021*; *Lipowski & Lipowska, 2012*). |

**Table 2  The advantages of ACO-GA hybrid method.**

| Algorithm | Aspect | Advantages |
|---|---|---|
| ACO | Initial solution | GenACO used the initial population by ACO which was not created randomly but followed pheromone trail based on a particular objective function. |
| | Solution probability | GenACO does not use a roulette wheel to select a solution from each ant. ACO needs to guarantee as many (near) optimum solutions as possible due to the fact that is just being run for the first time. The use of r0 will be used here. |
| GA | Avoiding local optimum | GA is more advantageous in the mechanism of selection, crossover, and mutation which create a reliable algorithm performance to avoid optimal local traps. |
| | Elitism | GenACO implements the elitism principle of the GA algorithm to improve the solution in each subsequent iteration. |

# METHODOLOGY

## GenACO

GenACO is an optimization framework that we propose to improve the performance of CGACA produced by *Wang et al. (2019)*. GenACO was proposed using the constant r0 to overcome the roulette wheel weakness in solution probabilities. The r0 value is used to balance exploring candidate solutions in the early iterations and focuses on exploiting the optimum solution at the end of the iteration. In addition, GenACO has two execution modes, namely cyclic and non-cyclic. Both managed to improve the quality of the solution.

Table 2 describes the rationale of GenACO maintaining the cyclic ACO-GA hybrid method. Figure 6 illustrates the step of the proposed hybrid GenACO. The ACO algorithm is needed to generate the best initial population for GA. The r0 has been used in this step. If

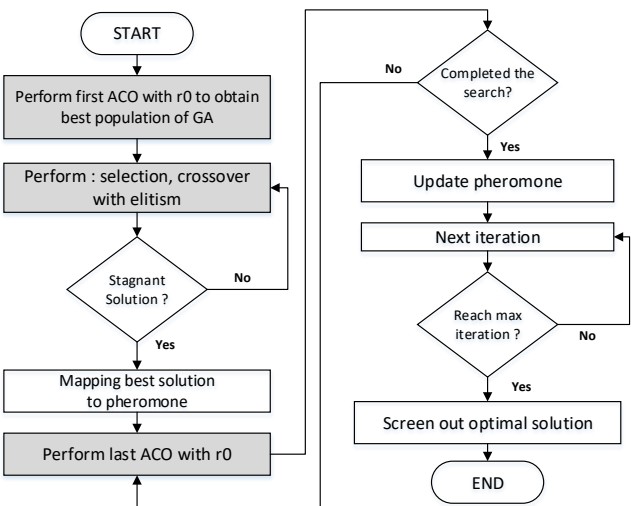

**Figure 6** The step of the proposed hybrid GenACO.

the GA has a stagnant solution, the last ACO will run again until the iteration ends. Detail explanation about r0 will be explained further in the next section.

The Knapsack problem is a classic optimization problem that can be solved using brute force or greedy methods. However, the greedy method does not guarantee an optimal solution, while the brute force has a very high time complexity of $O(2^n)$. Based on Fig. 6, assuming n is cached data, m is the number of GenACO ants, and Nc is the iteration used, then the complexity of GenACO is $O(Nc*n*m)$. Consequently, the time complexity of GenACO is better than the classical method.

### Solution probability

The ACO algorithm plays an essential role in generating the solution required by the GA to produce the best population. However, the disproportionate use of trace pheromones can cause the best solution of Eq. (4) not always to appear. The ACO algorithm in GenACO no longer uses the roulette wheel to select a solution from all the ants. It was replaced by adding r0 as a constant to be compared with a random number r [0,1], such that when r0 > r, the ant selects the cached data with the enormous pheromone value and selects the cached data randomly when otherwise.

$$\tau_i(t) = w_1 * \frac{ct(i)}{total(ct)} + w_2 * \frac{fr(i)}{total(fr)} + w_3 * \frac{sz(i)}{total(sz)} \qquad (2)$$

The initialization of the pheromone value ($\tau$) follows Eq. (2) by using three cached data property values, which include the access count (ct), access time (fr), and data size (sz) with each multiplied by its respective weight $w1 = 0.3$ $w2 = 0.3$ $w3 = 0.4$ *Wang et al. (2019)*. Moreover, the visibility function in Eq. (3) also influences the choice of solutions made by the ant colony. Therefore, the probability of cached data is selected and entered into the

cache server follows Eq. (4).

$$\eta_i(t) = w_1 * \frac{ct(i) * fr(i)}{sz(i)} \tag{3}$$

$$P_i(t) = \frac{\tau_i(t)^\alpha * \eta_i(t)^\beta}{\tau_i(t)^\alpha * \eta_i(t)^\beta + \tau_i(t)^\alpha * \eta_i(t)^\beta} \tag{4}$$

The r0 constant value becomes very important in determining the direction of ACO algorithm solution selection in GenACO. The simulation showed the r0 = 0.5 is suitable for placing in the first iteration of ACO to make the GenACO explore as many potential solutions as possible and not easily trapped on the local optimum in the initial iterations. In addition, r0 = 0.9 is planned to be set in the second part of ACO when the GA experiences a solution stagnation. This part is expected to focus more on exploiting the search for the optimum value. Based on experience, it will be difficult for the random number r[0,1] to be greater than 0.9. Therefore setting this value seems to force ACO always to choose the cached data with the most significant probability. The use of r0 in GenACO is illustrated in Table 3.

### Cyclic and non-cyclic GenACO
GenACO proposed two modes, cyclic and non-cyclic, to improve CGACA performance. The first mode is described in Table 3. In contrast, the second mode only runs the ACO algorithm once in the first iteration, after which the GA algorithm continues fully up to the maximum iteration. The non-cyclic mode does not use r0 = 0.9 to avoid the local optimum in the initial iteration. Therefore we used r0 = 0.7, which is expected to balance the exploration and exploitation of solutions in the first iteration, as illustrated in Table 4.

### Cached-data representation
The ACO and GA algorithms have different solution vector representations in modeling cached data. In the GA algorithm, a chromosome creates as much array space as the total data, with each representing the 0/1 condition of one cached data. A value of 1 indicates that the selected cached data will be entered into the cache server. At the same time, 0 means it will not be included. Meanwhile, in the ACO algorithm, the array space only consists of several routes selected by the ants. One ant and another may have a different number of selected routes.

Moreover, the route chosen by this ant represents a collection of cached data to be entered into the cache server, and the ACO solution representation on GenACO duplicates how ACO works in solving TSP. Figures 7 and 8 illustrate an example solution of GA and ACO, respectively.

### Multi-objective function
GenACO can solve the cached data offloading optimization problem using the knapsack problem approach as a multi-objective optimization model. Its multi-objective optimization is built using three property values which include access count (ct), access time (fr), and data size (sz) contained in each cached data. However, not all data can enter

**Table 3   Illustration on the use of r0 in the three cyclic GenACO steps.**

| | | | |
|---|---|---|---|
| **1** | First ACO | r0 = 0.5 | condition: if r0 > random r |
| | | | solution: random ($P_i(t)$)                    (5) |
| | | | assumption: it is easier for the random number r to be greater than 0.5 |
| **2** | | | Genetic Algorithm |
| **3** | Last ACO | r0 = 0.9 | condition: if r0 > random r |
| | | | solution: maximum ($P_i(t)$)                    (6) |

**Pseudo-code**

```
Input: ant m, iteration ith, cached_data n, visibility_matrix v(n),
probability P(n,v), server_capacity S
Output: Knapsack_items Kp, Obj_Func Fx

% initialization
foreach n=1: all cached_data do
     v(n) = eq.(3)
end for

while ith < ith_max do
  while S(Kp) < S_max do
    if ith == 1 then
    % run first ACO (Table 3: step-1)
       foreach m=1: all ants do
          foreach n=1: all cached_data do
             calculate P(n,v); generated random r[0,1]; r0=0.5;
             if (r0 > r) Kp(n) = eq.(5); calculate S(Kp); calculate Fx(S);
          end for
       end for
    end if
    if ith > 1 then
    %run GA (Table 3: step-2)
       Run crossover and mutation (Table 5)
       calculate profit(Kp); Update S(Kp); update Fx(S);
       if GA has stagnan then
          %run last ACO (Table 3: step-3)
          foreach m=1: all ants do
             foreach n=1: all cached_data do
                calculate P(n,v); generated random r[0,1]; r0=0.9;
                if (r0 > r) Kp(n) = eq.(6); Update S(Kp); update Fx(S);
             end for
          end for
       end if
    end if
  end
  screenout(Kp); screenout(Fx);
end
```

the cache server due to the limited capacity. Therefore, their priority value was determined based on these three property values.

These three variables also have their respective objective functions. The objective function of access count is in line with Eq. (8); access time with Eq. (9); and data size with Eq. (10), *Wang et al. (2019)*, multiplied by their respective weights $w_1 = w_2 = 0.3$, and $w_3 = 0.4$, which is then calculated the final objective function as profit (Fx) following (Eq. 11), *Wang et al. (2019)*.

The smaller Fx indicates high priority. The $n$ dataset consists of cached data candidates of $x_1, x_2, x_3, \ldots, x_j$ that can be entered by the cache server with capacity $S$ and profit $F_x$. Meanwhile, each $x_j$ consists of access count (ct), access time (fr), and data size (sz). Furthermore, the GenACO is expected to determine the solution vector $x_j$ when $x_j = 1$ to calculate objective function. The cached data is ignored when $x_j = 0$ and will not be

**Table 4   Illustration on the use of r0 in the two GenACO non-cyclic steps.**

| 1 | First ACO | r0 = 0.7 | condition: if r0 > random r |
| | | | solution: random $(P_i(t))$ and maximum $(P_i(t))$ |
| | | | assumption: a more balanced process of exploring and exploiting solutions |
| 2 | | | Genetic Algorithm |

**Pseudo-code**

```
Input: ant m, iteration ith, cached_data n, visibility_matrix v(n),
probability P(n,v), server_capacity S
Output: Knapsack_items Kp, Obj_Func Fx

% initialization
foreach n=1: all cached_data do
    v(n) = eq.(3)
end for

while ith < ith_max do
  while S(Kp) < S_max do
    if ith == 1 then
    % run first ACO (Table 3: step-1)
       foreach m=1: all ants do
          foreach n=1: all cached_data do
             calculate P(n,v); generated random r[0,1]; r0=0.7;
             if (r0 > r) Kp(n) = eq.(5);
             else Kp(n) = eq.(6);
             calculate S(Kp); calculate Fx(S);
          end for
       end for
    end if
    if ith > 1 then
    %run GA (Table 3: step-2)
       Run crossover and mutation (Table 5)
       calculate profit(Kp); Update S(Kp); update Fx(S);
    end if
  end
  screenout(Kp); screenout(Fx);
end
```

included in the objective function calculation following Eq. (11). Therefore problem definition following Eq. (7).

$$\sum_{j=1}^{n} profit(F_x)x_j \leq S \tag{7}$$

$$f_{CT} = \frac{1}{n}\sum_{j=1}^{n} \frac{D_{ct_{(j)}} - D_{ct_{(min)}}}{D_{ct_{(max)}} - D_{ct_{(min)}}} \tag{8}$$

$$f_{FR} = \frac{1}{n}\sum_{j=1}^{n} \frac{D_{fr_{(j)}} - D_{fr_{(min)}}}{D_{fr_{(max)}} - D_{fr_{(min)}}} \tag{9}$$

$$f_{SZ} = \frac{1}{n}\sum_{j=1}^{n} \frac{D_{sz_{(j)}} - D_{sz_{(min)}}}{D_{sz_{(max)}} - D_{sz_{(min)}}} \tag{10}$$

$$F_x = w_1 * (1 - f_{CT}) + w_2 * (1 - f_{FR}) + w_3 * f_{SZ} \tag{11}$$

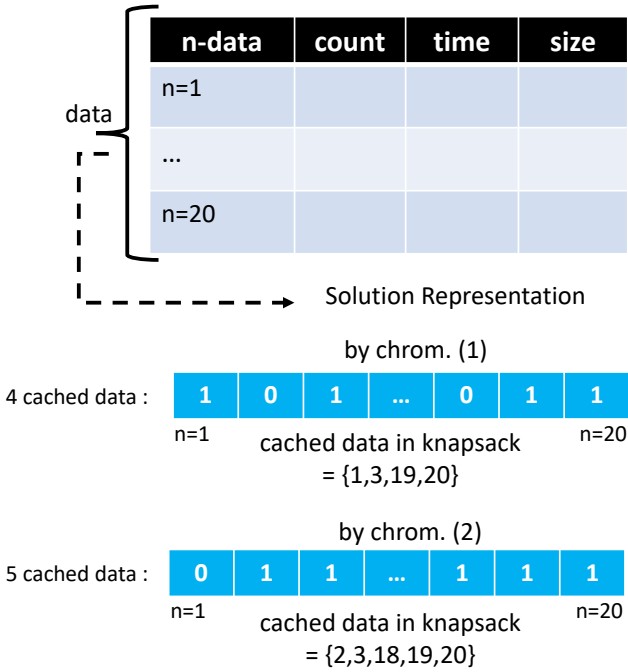

**Figure 7** An example of solution vector in GA algorithm.

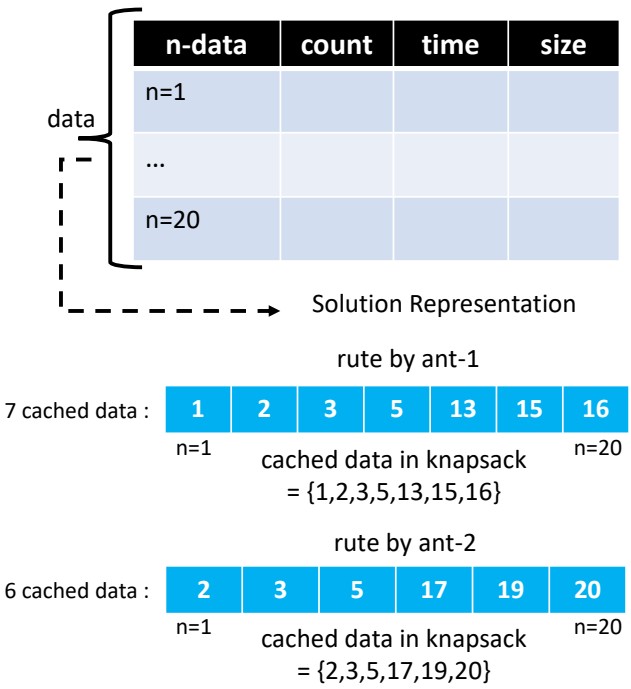

**Figure 8** An example of solution vector in ACO algorithm.

## Performance measurement

Evolutionary algorithms such as ACO and GA are widely used to complete optimization work. Generally, the optimization performance measurement through the continuous domain is usually measured by the optimum value of the objective function and the solution convergence time achieved (*Sahu, Panigrahi & Pattnaik, 2012*; *Sun, Song & Chen, 2019*). Meanwhile, the knapsack problem belongs to a discrete domain. Therefore, additional performance indicators are the best and worst profit, the average profit achieved, and the total number of items in the knapsack (*Rizk-Allah & Hassanien, 2018*; *Li et al., 2020a*; *Li et al., 2020b*; *Liu, 2020*). However, the best solution in the cached data offloading optimization case study was selected based on the highest number of knapsack items with the lowest objective function value.

# EXPERIMENTS AND EVALUATION

## Dataset

GenACO was tested by datasets used in *Wang et al. (2019)* with three cached data property: access count (ct), access time (fr), and data size (sz) with their respective sizes in Mbit, decimal, and second. Moreover, the maximum capacity of the cache server was 950 Mbit. The dataset can be accessed through the given GitHub repository.

## Simulation setup

The single ACO and GA scenario was divided into four scenarios. Moreover, the CGACA scenario was repeated three times to observe the solution stagnation position experienced by GA. The results were compared with the proposed cyclic and non-cyclic GenACO. The simulation was conducted using 10 particles and 20 iterations with the parameter settings of the six scenarios presented in Table 5. Both parameters were adopted from *Wang et al. (2019)* then simplified because the best solution average has been found in such particles and iterations range. The simulation was conducted using PHP programming language and Mysql database to facilitate our subsequent research in implementing fog computing architecture.

## Results and discussion
### *Performance comparison of single ACO*

The complete simulation results in this paper can be seen in the Tables S1–S4. Table S1 showed the comparison of cached data optimization results for the single ACO algorithm. According to *Wang et al. (2019)*, the best known objective function (Fx) for cached data offloading is 0.4374 with an optimum cached data (ch.dt) of 15. The first part of Table S1 showed the ACO algorithm from the CGACA section, and scenario-1a was observed to have relied on the roulette wheel on the cached data selection mechanism to be loaded in cache storage. However, this roulette wheel mechanism has two weaknesses (i) produce random solutions (ii) cause the emergence of superior individuals. The average of Fx and cached data generated in scenario-1a are the worst. The solution looks random and lacks convergence

Scenarios: 1b, 1c, and 1d are single ACO obtained from parts of the GenACO framework and observed improvement in solution probability as described in 'GenACO'. The solution

**Table 5  Parameters settings.**

| Scenario | Algorithms | Parameters |
|---|---|---|
| 1 | Single ACO (CGACA) | a) $\alpha = 0.3; \beta = 0.5; \rho = 0.001$ |
| | Single ACO (GenACO) | b) $\alpha = 0.3; \beta = 0.5; r_0 = 0.3; \rho = 0.001$ |
| | | c) $\alpha = 0.3; \beta = 0.5; r_0 = 0.5; \rho = 0.001$ |
| | | d) $\alpha = 0.3; \beta = 0.5; r_0 = 0.7; \rho = 0.001$ |
| 2 | Single GA | a) prob.mut.= 0.12; crossover (P1 = 15%; P2 = 85%) |
| | | b) prob.mut.= 0.25; crossover (P1 = 15%; P2 = 85%) |
| | | c) prob.mut.= 0.25; crossover (P1 = 35%; P2 = 65%) |
| | | d) prob.mut.= 0.5; crossover (P1 = 50%; P2 = 50%) |
| 3 | CGACA | ✓ First ACO: $\alpha = 0.3; \beta = 0.5; \rho = 0.001;$ ✓ GA: mutation = *random*; crossover = *roulette wheel* ✓ Last ACO: $\alpha = 0.3; \beta = 0.5; \rho = 0.001;$ |
| 4 | Cyclic GenACO | ✓ First ACO: $\alpha = 0.3; \beta = 0.5; r_0 = 0.7; \rho = 0,001$ ✓ GA: Mutation probability = 0,12; crossover = P1:15%, P2:85% ✓ Last ACO: $\alpha = 0.3; \beta = 0.5; r_0 = 0.9; \rho = 0.001$ |
| 5 | Non cyclic GenACO | ✓ First ACO: $\alpha = 0.3; \beta = 0.5; r_0 = 0.7; \rho = 0.001$ ✓ GA: Mutation probability = 0,25; crossover = P1:15%, P2:85% |

probability based on r0 was compared with a random number r[0,1]. If r0 > r, then the solution will follow (6), otherwise it will follow (5). Scenario-1b was set using r0 = 0.3 based on the assumption that the random number r will be easier to be greater than this value, and the scenario is expected to have more variety of solutions. However, scenario-1b managed to obtain an optimal solution of 15 cached 7 times out of 20 iterations. The exciting thing from these results is that the optimal solution was generated from different Fx values, 0.4374, 0.4405, and 0.4382. The best solution was the smallest was 0.4374.

Scenario-1c was set using r0 = 0.5 and managed to obtain an optimal solution of 15 cached 5 times out of 20 iterations. Initially, r0 = 0.5 was assumed to have the capacity to produce more optimal cached data than r0 = 0.3, but it did not. It proves unreliable to have the random r value in a position more than or less than the r0 value. Meanwhile, the five best optimal solutions found in this scenario have the same Fx value of 0.4382, and this means all the 15 selected cached data have the same id_data. In this case, the best solution in scenario-1b is better than scenario-1c due to its smaller Fx value.

The last single ACO, scenario-1d, has the highest value of 0.7 than the previous two scenarios. The solution selection is expected to lead more to equation (6). Meanwhile, Table S1 showed this scenario-1d produces the best solution compared to the previous two scenarios by having 95% optimal solution with 19 out of the 20 iterations having 15 cached data with two dominant Fx values, which are 0.435 and 0.4405. It is important to note that the Fx value of 0.435 became the best value for the cached data offloading

objective function generated from the overall single ACO simulation scenario. The results of this simulation are in accordance with our manual calculations to improve the results of previous research conducted by *Wang et al. (2019)*.

### Performance comparison of single GA

Table S2 compared the single GA algorithm with four different mutation probability scenarios and crossover compositions. The first part of the single GA simulation was observed to have used a crossover composition of Parent-1 (P1) = 15% and Parent-2 (P2) = 85%. It means P1 gave the first gene to the third gene, and P2 gave the fourth gene to the twentieth gene when the crossover occurred. This calculation was used in all crossover scenarios in this single GA simulation. Another critical parameter is gene mutation probability. The probability of gene mutation in the simulation was 0.5, expecting that the crossover mechanism will not create a new fitness value smaller than the previous best value.

Scenario-2a showed the worst solution among all the single GA simulation scenarios, with the best solution produced having only 11 cached data with the same Fx values of 0.4653. At first glance, the solution seems to be rapidly converging but quickly trapped in the local optimum due to the inability of the crossover mechanism to produce a variety of solutions. Meanwhile, scenario-2b managed to determine a (near) optimal solution with 14 cached data starting from the 13th to the maximum iteration, and the search for solutions was varied. Moreover, its higher mutation probability value compared with the previous scenario caused each gene in the chromosome to have more flexibility to improve the solution quality in the next generation.

Scenario-2c uses the same mutation probability value of 0.25 as scenario-2b but has a crossover composition of P1 and P2, slightly different from scenario-2b. Therefore, the results also differ significantly from the average Fx value and the cached data produced. The last scenario-2d used a mutation probability value of 0.5, with P1 and P2 having 50% composition each. This scenario is generally similar to scenario-2a by being a fast solution towards convergence, but it was quickly trapped at the optimum local value at the beginning of the iteration. Moreover, the solution improvement with a mutation probability value of 0.5 also failed to show, and this means the parameter setting in GA greatly affected the quality of the solution produced.

Table S2 has shown that some of the best solutions produced by a single GA seem to be more optimal (with a smaller Fx) than the best known Fx. However, this smaller Fx value does not indicate a better solution because the cached data (ch.dt) is less than the best-known cached data. Based on the best solution to the knapsack problem approach shown in Eq. (7), the single GA solution has not outperformed the best-known solution.

The cached data offloading solution produced by GA is no better than ACO. None of the solutions from GA achieved the best-known value. GA does not guarantee the search for a (near) optimum solution as formulated by the pheromone function in ACO. However, GA has a significant role in accelerating the convergence of cached data offloading case study solutions. If correlated between Fig. 6 Tables S3 and S4, ACO and GA have a vital function in their respective positions. In scenario-1b and scenario-1d, the ACO algorithm can find

the optimal solution in less than five iterations. This initial solution from ACO can be used as an outstanding initial population for the GA algorithm to continue the solution search process until the maximum iteration. This is supported by Tables S3 and S4, which show the trend of solution improvement from beginning to end.

### Performance comparison of CGACA

Table S3 showed the simulation results for the CGACA framework with the ACO algorithm run in the first iteration to generate the initial population. At the same time, the GA continued the process of determining the solution. However, the GA algorithm was expected to experience solution stagnation, which led to the application of the ACO algorithm for the second time to continue the solution search process up to the maximum iteration. In the first part, the GA solution was observed to have stagnated in the fifth iteration, and the ACO continued the search process from the sixth iteration to the maximum. Therefore, this part of the experiment was the worst case of CGACA since it required the longest execution time. It was associated with the quick stagnation of GA, which led to the use of ACO to complete the solution search process until the maximum iteration was reached.

In the second part, the GA algorithm did not experience any solution stagnation. Therefore, the required execution time was the best, but it could not determine an optimal solution since the initial population created by ACO was not optimal. This is one of the disadvantages of the roulette wheel. It is also important to note that selecting a solution based on the roulette wheel was only in one iteration, and the optimal solution was not found. Moreover, the GA algorithm has a solution stagnation at the end of the iteration, precisely in the 13th iteration. It means the time consumption at the end of the CGACA simulation is not as much as the first part.

Meanwhile, the quality of the CGACA solution in the third part was the best, and the solution improvement on the last-ACO was expected to be successfully conducted by producing a (near) optimum solution of 14 cached data. The same was found in the first part of CGACA, but the Fx value produced in the third part is better. Furthermore, the simulation results in and Table S3, showed that the CGACA total time consumption increased along with the number of iterations completed by the ACO. A more significant portion of iterations run by ACO led to the generation of more extended time consumption. It means ACO required a long execution time to update the pheromone trace on each selected cached data by all ants in each iteration. Therefore, it is very inefficient if it has to complete the solution search by a single ACO.

### Comparison with cyclic and non-cyclic GenACO

Tables S3 and S4 marks the solution from ACO with a dark color on the cell background. Table S4 compared the performance of cached data offloading using the CGACA and GenACO frameworks. The GenACO simulations were divided into cyclic and non-cyclic scenarios. Both utilized ACO to run once in the first iteration while the GA continue the process and the last ACO was prepared to help when the GA experiences a solution stagnation.

The first part showed the simulation result obtained from the CGACA framework, which was observed to have succeeded in determining the optimum solution based on the best value of 15 cached data with Fx = 0.4374, but this was only found once in the 18th iteration. The first part of Tables S4 showed that the GA stagnated at the 7th iteration, leading to the last ACO from the 8th iteration to the maximum. In the end, CGACA required more than 40 s which is not too different from the value in Tables S3, and this higher total time was associated with the dominance of execution by ACO.

The second part of Tables S4 showed the results for the GenACO simulation with cyclic mode, and the working principle was found to be precisely the same as CGACA (baseline). However, the probability of a cached data selection solution was improved. The first ACO on cyclic GenACO was set using r0 = 0.7 with the expectation of generating more (near) optimal solutions. The results showed that the GA algorithm has succeeded in determining the optimum solution of 15 cached data with an Fx value of 0.4350 better than the best-known Fx value between the 2nd and 12th iterations. The improvement is only 0.0024 but based on the raw data we tracked, GenACO can accommodate more small cached data. The server utility is maximized because it can enter more cached data in the future.

However, the solution produced from the 13th iteration was considered stagnant since the Fx value is the exact five times in a row. It led to the application of the last ACO starting from the 14th up to the maximum iteration. The solution produced did not get better. It was discovered that all ants could not determine the optimal solution previously obtained until the maximum iteration was run. It means the last ACO was stuck at the local optimum because the 14 cached data solutions added more pheromone concentration than the previous one, the 15 cached data solutions. Therefore, the last ACO preferred to follow the solution of the 14 cached data. It was influenced by setting r0 = 0.9 on the last ACO, which directed the ants to the path with the strongest pheromone than the new path required to be explored.

The third part of Tables S4 showed the results of the non-cyclic GenACO simulation, which only ran ACO on the first iteration and the GA on the second iteration up to the end. The GA was observed to have stagnated solutions for five consecutive iterations but continuously searched for solutions up to the maximum iteration. Meanwhile, r0 = 0.9 was used because ACO was used only once. The ants preferred the cached data with maximum probability, and several (near) optimum solutions are expected to be obtained by ACO ants in this non-cyclic mode to form the best initial population in the GA algorithm. The 3rd to the 5th iteration results showed that the GA immediately found the optimal solution for the best known 15 cached data with Fx = 0.4350 and maintained this value convergently from the 10th to the maximum iteration. Based on Table 6, the proposed hybrid GenACO method, particularly non-cyclic mode, is superior to single ACO, GA, and CGACA in obtaining an average Fx, best solution, the optimal amount of cached data during iteration. Full results can be seen in the Supplemental Information.

**Table 6  A comparison of the best results from single GA, ACO, CGACA, and GenACO.**

| | Best scenario comparison | | | | | | | |
|---|---|---|---|---|---|---|---|---|
| | Single GA | | Single ACO | | CGACA | | GenACO | |
| | Fx | ch.dt | Fx | ch.dt | Fx | ch.dt | Fx | ch.dt |
| AVG | 0.4012 | 12.95 | 0.4358 | 14.95 | 0.4851 | 12.75 | 0.4342 | 14.71 |
| BEST | 0.3855 | 14 | 0.435 | 15 | 0.4476 | 14 | 0.435 | 15 |
| WORST | 0.4105 | 10 | 0.4305 | 14 | 0.4749 | 11 | 0.4305 | 14 |

**Table 7  Solution comparison of different r0.**

| | Comparing the r0 | | | | | | | |
|---|---|---|---|---|---|---|---|---|
| Try | r0=0.3 | | r0=0.5 | | r0=0.7 | | r0=0.9 | |
| | Fx | ch.dt | Fx | ch.dt | Fx | ch.dt | Fx | ch.dt |
| 1 | 0.4947 | 10 | 0.4408 | 12 | 0.4470 | 14 | 0.4355 | 13 |
| 2 | 0.4894 | 12 | 0.4727 | 12 | 0.4355 | 14 | 0.4355 | 14 |
| 3 | 0.4856 | 11 | 0.4575 | 13 | 0.4498 | 12 | 0.4355 | 14 |

### The impact of different r0

Based on Table 7, the value of r0 has an essential role in producing a solution used as an initial parent for the GA algorithm. At $r0 = 0.3$, the solutions produced by ACO vary widely. Three experiments conducted at $r0 = 0.3$ resulted in three different Fx and ch.dt values. This is in line with the first ACO goal in the GenACO framework, which is to focus on solution exploration. However, the resulting solution with $r0 = 0.3$ is quite far from the target best-known value, so that this will make GA work difficult. The results shown are also in the use of $r0 = 0.5$. However, the quality of the solution of $r0 = 0.5$ is better than $r0 = 0.3$. In general, the two values of r0 are still not close to the best-known value, and the resulting solution does not appear to be convergent. This proved our hypothesis in the Solution Probability section: the smaller the value of r0, the more the solution chosen by the ant colony refers to equation (5). The use of $r0 = 0.7$ and $r0 = 0.9$ resulted in better solution quality. Both r0 solutions begin to look convergent. The Fx and ch.dt values are getting closer to the best-known values. But keep in mind that $r0 = 0.7$ or $r0 = 0.9$ is not recommended for use in the first ACO because it can reduce the opportunities for solution exploration. Both are better used in the last ACO to narrow the search space to focus on exploiting the best value towards the best-known. The impact of using r0 on the GenACO framework can be seen again, as shown in Tables S4. Based on Tables S4, the ACO solution using $r0 = 0.7$ helps GA find the optimum solution and achieve solution convergence.

### Comparison of stagnant solution CGACA and GenACO

Figure 9 showed the solution stagnation of the three ACO–GA hybrid simulations. The first ACO on CGACA obtains an initial solution that is not very good. This situation impacts GA performance which fails to improve the initial solution so that it is quickly trapped in the local optimum. Moreover, the roulette wheel mechanism and the probability of crossover and mutation cause GA to fail to improve this initial solution. In the end, GA

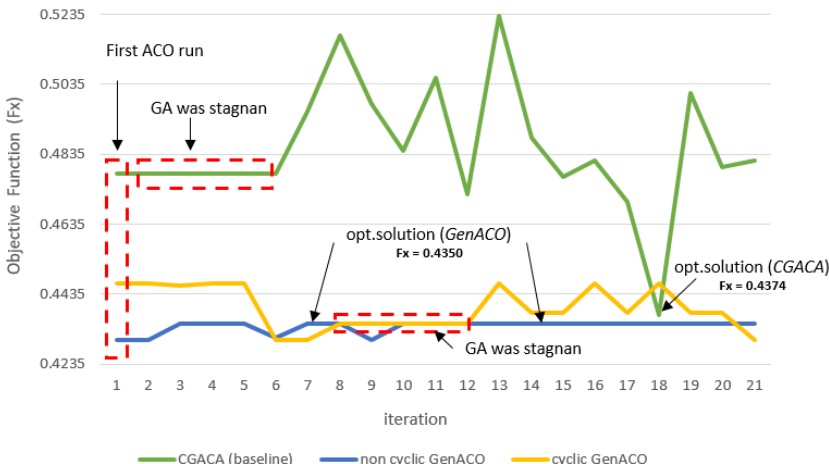

**Figure 9  Objective function (Fx) comparison between CGACA and GenACO.**

stuck to the solution stagnation from iterations 2 to 5. The GA algorithm, which stagnated the solution far from the best known, caused the last ACO hard to improve the solution and achieve solution convergence. However, the roulette wheel mechanism used by the last ACO at CGACA failed to fix this until the maximum iteration.

The GA algorithm on Cyclic GenACO also suffers from a solution stagnation. However, before solution stagnation occurred, GA had a better initial solution. GA successfully maximized the initial solution of the first ACO to become an optimal solution. However, because the optimal solution is equal in five consecutive iterations, it is considered a stagnation of the solution. In the end, the last ACO on cyclic GenACO was executed, but the optimal solution previously generated could not be maintained. We suspect this situation was caused by set r0 = 0.9, so the solution fell out from the previous optimum. In future work, this will be our concern so r0 in the last ACO can be more adaptive to the previous solution.

The use of r0 = 0.7 in the non-cyclic GenACO simulation succeeded in obtaining a better initial solution than CGACA and cyclic GenACO. This situation makes GA job easier. In addition, the proper parameter setting on the probability of mutation and crossover makes GA performance more reliable. It can be seen in the performance of GA, which can maintain the optimal solution until the maximum iteration.

Based on Fig. 9, non-cyclic GenACO is the best solution for average solution quality and total time consumption. However, GenACO cyclic and non-cyclic modes did not have a significant difference in total time consumption. Non-cyclic GenACO is only 4.5 s ahead of cyclic mode. Based on Tables S4, the saving time obtained by GenACO can be calculated using Eq. (12).

$$Saving_{time} = \frac{(CGACA\ exec.time - GenACO\ exec.time)}{CGACA\ exec.time} * 100\%. \tag{12}$$

Therefore, we calculated cyclic GenACO saving time consumption by up to 38%, while non-cyclic GenACO saving it by 47%. Both GenACO modes can be accepted as a solution

to the problem of cached data offloading. The ACO algorithm is very appropriate to use in the first iteration to create an excellent and reliable initial population, making it easier for GA to find the optimal solution quickly and good average quality of the solution.

## CONCLUSIONS

An edge computing framework must have a cached data offloading capability that needs to be optimized due to the limited storage capacity and maximum possible profit. This paper proposed improving the hybrid ACO-GA algorithm performance using a new GenACO framework with cyclic and non-cyclic modes. The simulation results show that GenACO minimizes the objective function of cached data optimization from 0.4374 to 0.4350. In addition, GenACO can also outperform the quality of the solution in terms of the average objective function generated. Moreover, the parameter setting in evolutionary algorithms plays an essential role in the overall algorithm performance.

Based on the simulation results, non-cyclic GenACO is the best mode for solving cached data offloading optimization. This mode can reduce time consumption by up to 47%. Our further research tests GenACO by measuring the hit ratio, examining the impact of latency and response time on the user side in an edge computing environment.

## ACKNOWLEDGEMENTS

The authors thank the supervisory team for their suggestions and motivation.

### Funding

This research was funded by the Final Project Recognition (RTA) Grant No. 3190/UN1/DITLIT/DIT-LIT/PT/2021 from Gadjah Mada University (UGM). LPDP provided a scholarship through the BUDI-DN in the Electrical Engineering Doctoral Study Program. The funders had no role in study design, data collection and analysis, decision to publish, or preparation of the manuscript.

### Grant Disclosures

The following grant information was disclosed by the authors:
The Final Project Recognition (RTA) from Gadjah Mada University (UGM): 3190/UN1/DITLIT/DIT-LIT/PT/2021.
LPDP provided a scholarship through the BUDI-DN in the Electrical Engineering Doctoral Study Program.

### Competing Interests

The authors declare there are no competing interests.

### Author Contributions

- Mulki Indana Zulfa conceived and designed the experiments, performed the experiments, analyzed the data, performed the computation work, prepared figures and/or tables, authored or reviewed drafts of the paper, and approved the final draft.

- Rudy Hartanto and Waleed Ali conceived and designed the experiments, performed the experiments, analyzed the data, prepared figures and/or tables, authored or reviewed drafts of the paper, and approved the final draft.
- Adhistya Erna Permanasari conceived and designed the experiments, performed the experiments, authored or reviewed drafts of the paper, and approved the final draft.

## Data Availability

The data is available at GitHub: https://github.com/mulkiiz/phd/tree/master/GenACO.

## Supplemental Information

Supplemental information for this article can be found online at http://dx.doi.org/10.7717/peerj-cs.729#supplemental-information.

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
