# Peer review of "GenACO a multi-objective cached data offloading optimization based on genetic algorithm and ant colony optimization"

_PeerJ Computer Science, doi:10.7717/peerj-cs.729_

## Round 0.1 · original submission · Major Revisions

The article is interesting, but I think it needs to be considered in more detail in order to understand the subject.

Reviewer 1 ·

Basic reporting

The paper presents an interesting multi-objective cached data offloading optimization approach.

Experimental design

Following issues needs to be addressed.
1. Pls. format your paper according to the journal guideline.
2. Quantify your results in the abstract
3. Add a gap analysis section in your introduction and also mention the research questions.
4. Fig. 1 is generic, plz add steps specific to your solution.
5. Discussion section is missing.

Validity of the findings

As mentioned above

Additional comments

As mentioned above

Reviewer 2 ·

Basic reporting

- Abstract, row 26 “GenACO” and row 38 “ACO-GA”; every non-usual acronym (for the general reader) must be explained at its first use. The extended versions of these abbreviations must be written at their first uses.
- Row 73, “Z={3, 4,5} or Z={2.5}”. Correct: “Z={3,4,5} or Z={2,5}”
- Row 78, “However, this method still leaves problems..”. Correct: “However, these methods still leave problems..”
- Row 88, “resulting solution not converge quickly.”. Correct: “resulting solution does not converge quickly.”
- Row 90, “CGACA”; every non-usual acronym (for the general reader) must be explained at its first use. The extended version of this abbreviation must be written at its first use.
- Row 142, “Mobile Edge Computing (MEC)”. Correct: “MEC”
- Row 227, “r0”. Correct: “r0”, It should be “r0” in the rest of the text
- Row 250, “ct, fr, sz”. Correct: “ct, fr, sz”. These notations must be used in all formulas and in the rest of the text.
- Row 251, “w1, w2, and w3 “. Correct: “w1, w2, and w3 “. Correct notations should be used in the rest of the text
- Row 302, “Fx”. Correct: “Fx”.This notation should be used in the rest of the text and formulas.
- Row 339, “Appendix at the of of this”. Correct: “Appendix at the end of this”
- Row 356, “scenarios-1b”. Correct: “scenario-1b”
- Row 374, “ conducted by Wang et al.”. Correct: “conducted by Wang et al.(years must be written)”
- Row 452, “Table 9 compares…”; Row 459 “..in Table 9 showed the..”. This is the same for all table-related expressions. There must be time consistency. Simple present tense or simple past tense?
- Row 471, “..five times in a row and this led to..”. Correct: “..five times in a row, and this led to..”
- Row 472, “..to the maximum iteration but..”. Correct: “..to the maximum iteration, but..”
- Row 491, “Based on table 10,”. Correct: “Based on Table 10,”
- Row 536, “algorithms plays an important..”. Correct: “algorithms play an important..”
- “Table 5 Parameters setting”. Correct: “Table 5 Parameters settings”
- “Table 10 A comparison of the best result from single GA, ACO, CGACA, and GenACO”. Correct: “Table 10 A comparison of the best results from single GA, ACO, CGACA, and GenACO”.
- And other grammatical mistakes. Grammarly software and WORD spelling&grammar tool can be used.
- In general, the whole paper must be proofread by a very good English speaker (and writer).

Experimental design

Research question well defined, relevant & meaningful. It is stated how research fills an identified knowledge gap. Rigorous investigation performed to a high technical & ethical standard.

Validity of the findings

1) The r0 parameter was used in the study and it was stated that some r0 values gave better results. The authors should discuss the effect of different r0 values on the results in experimental studies by using specific ranges.
2) It is mentioned in the text that evaporation will be more on a long path and it is difficult for ants to follow this path. Therefore, In figure5, it is said that path b is shorter, but isn't path a shorter?
3) Authors should indicate the time complexity of the proposed algorithm. The degree of complexity of this problem should be described well. Otherwise, there is a perception that the problem can be solved with classical optimization problems.
6) When comparing the results obtained, the authors show that the objective function produces more appropriate solutions than other approaches. However, statistical (Friedman's test and Wilcoxon signed-rank test etc.) methods should be used to show whether the obtained results provide a significant improvement.
7) Since the proposed approach uses a single objective function, it should also be compared with the proposed state-of-the-art approaches (grey wolf optimizer, artificial algae optimization etc.) for binary optimization and discrete optimization methods.

Additional comments

-

Annotated reviews are not available for download in order to protect the identity of reviewers who chose to remain anonymous.

Reviewer 3 ·

Basic reporting

The study is about optimization of limited memory used for edge computing in the field of IoT. For this purpose, a new framework called GenACO is proposed to improve the performance of cached data optimization implemented in previous research. GenACO incorporates a solution selection probability mechanism to more reliably balance the discovery and exploitation process involved in finding solutions. Also, GenACO has two modes: cyclical and non-cyclical, it has been noted to have the ability to increase optimal cached data resolution, improve average solution quality, and reduce overall time consumption from previous research results. It has been noted that acyclic GenACO gives better results in solving cached data dump optimization.

Experimental design

The research question is about eliminating the disadvantages of the proposed ACO-GA method to solve the data dumping problem. These problems are that the execution time is quite long and the roulette wheel does not allow the solution to converge quickly. In addition, it is stated that the iteration constraints in the GA algorithm cause the GA performance to be non-optimal. For these reasons, a new hybrid method of the ACO-GA algorithm, named GenACO, framework is proposed to improve the performance of CGACA in this study.
In summary, the research question was well defined and aimed to solve an emerging problem in the field of IoT. The proposed solution for the problem includes the use of known methods as a hybrid. However, the algorithms of the proposed method are explained with unclear figures. Related methods should be made more descriptive by using pseudo code or block diagrams.
The studies conducted for the experiment and the datasets used here do not contain enough detail.

Validity of the findings

The simulation results presented in the study show that GenACO's cached data optimization reduced the objective function from 0.4374 to 0.4350. It has also been stated that the non-cyclical GenACO reduces time consumption by up to 47%. However, it is not mentioned how this result related to time was obtained.

Additional comments

In line 85, the word “Wang et al.” is written repeatedly.
The study is valuable in that it offers a solution to a current problem. However, the improvement achieved for memory space with the proposed method is very limited. The time-related gain, on the other hand, is not clearly expressed. The work is acceptable after the corrections stated in the other titles.

---

## Round 0.2 · accepted · Accept

The article can be accepted for publication as the revision requests of the referees are met by the authors.

Congratulations.

Reviewer 2 ·

Basic reporting

Necessary changes that I propose for the articles have been implemented.

Experimental design

Necessary changes that I propose for the articles have been implemented.

Validity of the findings

Necessary changes that I propose for the articles have been implemented.

Additional comments

Necessary changes that I propose for the articles have been implemented.

Reviewer 3 ·

Basic reporting

The study is about optimization of limited memory used for edge computing in the field of IoT. For this purpose, a new framework called GenACO is proposed to improve the performance of cached data optimization implemented in previous research. GenACO incorporates a solution selection probability mechanism to more reliably balance the discovery and exploitation process involved in finding solutions. Also, GenACO has two modes: cyclical and non-cyclical, it has been noted to have the ability to increase optimal cached data resolution, improve average solution quality, and reduce overall time consumption from previous research results. It has been noted that acyclic GenACO gives better results in solving cached data dump optimization.

Experimental design

The research question is about eliminating the disadvantages of the proposed ACO-GA method to solve the data dumping problem. These problems are that the execution time is quite long and the roulette wheel does not allow the solution to converge quickly. In addition, it is stated that the iteration constraints in the GA algorithm cause the GA performance to be non-optimal. For these reasons, a new hybrid method of the ACO-GA algorithm, named GenACO, framework is proposed to improve the performance of CGACA in this study.

In summary, the research question was well defined and aimed to solve an emerging problem in the field of IoT. The proposed solution for the problem includes the use of known methods as a hybrid.
The algorithms of the proposed method are explained with Pseudo-code. (This explanation was not clear)
The studies conducted for the experiment and the datasets used here did not contain enough detail. Authors have added an explanation.

Validity of the findings

The simulation results presented in the study show that GenACO's cached data optimization reduced the objective function from 0.4374 to 0.4350. It has also been stated that the non-cyclical GenACO reduces time consumption by up to 47%. However, it was not mentioned how this result related to time was obtained.
Authors has been added Eq.(12) as well as an explanation of how to get this 47% saving time on lines: 520-522.

Additional comments

The study is valuable in that it offers a solution to a current problem. However, the improvement achieved for memory space with the proposed method is very limited. The corrections mentioned in the previous review were carried out.
Although the gain of the study in terms of memory is limited, I think that the study is acceptable.